# Prevalence of Latent Tuberculosis Infection among Patients Undergoing Regular Hemodialysis in Disenfranchised Communities: A Multicenter Study during COVID-19 Pandemic

**DOI:** 10.3390/medicina59040654

**Published:** 2023-03-26

**Authors:** Mohamad Bachar Ismail, Nesrine Zarriaa, Marwan Osman, Safa Helfawi, Nabil Kabbara, Abdel Nasser Chatah, Ahmad Kamaleddine, Rashad Alameddine, Fouad Dabboussi, Monzer Hamze

**Affiliations:** 1Laboratoire Microbiologie, Santé et Environnement (LMSE), Doctoral School of Sciences and Technology, Faculty of Public Health, Lebanese University, Tripoli 1300, Lebanon; 2Faculty of Science, Lebanese University, Tripoli 1300, Lebanon; 3Cornell Atkinson Center for Sustainability, Cornell University, Ithaca, NY 14853, USA; 4Department of Public and Ecosystem Health, College of Veterinary Medicine, Cornell University, Ithaca, NY 14853, USA; 5Nini Hospital, Tripoli 1300, Lebanon; 6Dar Al-Chifae Hospital, Tripoli 1300, Lebanon; 7Orange Nassau Hospital, Tripoli 1300, Lebanon

**Keywords:** latent tuberculosis infection, hemodialysis, QuantiFERON-TB Gold Plus assay, Lebanon

## Abstract

*Background and Objectives*: Due to their weakened immune response, hemodialysis (HD) patients with latent tuberculosis infection (LTBI) are at higher risk for active tuberculosis (TB) disease and are more subject to patient-to-patient transmission within dialysis units. Consequently, current guidelines advocate screening these patients for LTBI. To our knowledge, the epidemiology of LTBI in HD patients has never been examined before in Lebanon. In this context, this study aimed to determine LTBI prevalence among patients undergoing regular HD in Northern Lebanon and to identify potential factors associated with this infection. Notably, the study was conducted during the COVID-19 pandemic, which is likely to have catastrophic effects on TB and increase the risk of mortality and hospitalization in HD patients. *Materials and Methods*: A multicenter cross-sectional study was carried out in three hospital dialysis units in Tripoli, North Lebanon. Blood samples and sociodemographic and clinical data were collected from 93 HD patients. To screen for LTBI, all patient samples underwent the fourth-generation QuantiFERON-TB Gold Plus assay (QFT-Plus). Multivariable logistic regression analysis was used to identify the predictors of LTBI status in HD patients. *Results*: Overall, 51 men and 42 women were enrolled. The mean age of the study population was 58.3 ± 12.4 years. Nine HD patients had indeterminate QFT-Plus results and were therefore excluded from subsequent statistical analysis. Among the remaining 84 participants with valid results, QFT-Plus was positive in 16 patients, showing a positivity prevalence of 19% (95% interval for *p*: 11.3%, 29.1%). Multivariable logistic regression analysis showed that LTBI was significantly associated with age [OR = 1.06; 95% CI = 1.01 to 1.13; *p* = 0.03] and a low-income level [OR = 9.29; 95% CI = 1.62 to 178; *p* = 0.04]. *Conclusion*: LTBI was found to be prevalent in one in five HD patients examined in our study. Therefore, effective TB control measures need to be implemented in this vulnerable population, with special attention to elderly patients with low socioeconomic status.

## 1. Introduction

Tuberculosis (TB), caused by the *Mycobacterium tuberculosis* complex (MTBC), is the world’s most common cause of death from a single infectious agent next to coronavirus disease 2019 (COVID-19) and one of the leading causes of death from antimicrobial resistance [1,2]. According to the World Health Organization (WHO), a quarter of the world’s population has TB infection, a minority of whom will rapidly develop active TB. However, the majority (>90%) of infected individuals develop effective acquired immunity to MTBC, which persists within the human host as latent TB infection (LTBI) [2]. Individuals with LTBI present a persistent immune response to stimulation by mycobacterial antigens without evidence of clinically manifested TB disease. However, they can potentially develop active TB, usually when their immune system becomes weakened. In 2021, there were an estimated 1.6 million deaths from TB worldwide [2]. Notably, COVID-19 is likely to have catastrophic effects on TB. Due to the COVID-19 pandemic, it was predicted that TB deaths in high-TB settings would rise by up to 20% over 5 years [3]. Similarly, TB disease can worsen COVID-19. Indeed, TB was found to be an independent risk factor for increased mortality due to COVID-19 [4,5]. Similarly, the risk of death and percentage recovery in COVID-19 patients with TB were, respectively, 2.17 times higher and 25% lower than in those without TB [6].

Chronic kidney disease (CKD) is a relatively neglected global public health threat, with high morbidity and mortality rates [7]. Indeed, CKD accounts for more than one million deaths yearly, which makes it the 12th leading cause of death worldwide [8]. Advanced renal impairment leads to end-stage renal disease (ESRD), necessitating renal replacement therapy, such as HD [9]. CKD impairs both innate and adaptive host immunity. The coexistence of chronic immune activation and chronic immune suppression is a common implication of uremia, resulting in a weak vaccination response and an increased incidence of cancers and microbial infections [10,11]. The TB reactivation risk is increased by 10–25 times among dialysis patients compared to the general population [12]. The mortality rate of TB in dialysis patients appears to be higher as well [13]. This may be commonly associated with late diagnosis and treatment, which, in turn, is due to the nonspecific symptoms of TB in dialysis patients, whereas the involvement is often extrapulmonary [13,14].

In this context, screening and treatment for LTBI in this population are recommended by the World Health Organization (WHO) to prevent the reactivation of LTBI and the secondary transmission of the infection within dialysis units [15]. While there are numerous available diagnostic assays to detect active TB, the diagnosis of LTBI remains challenging, relying on the detection of an immune response against MTBC antigens. A gold standard for the diagnosis of LTBI is lacking, but screening tests (tuberculin skin test (TST) and interferon-gamma release assays (IGRAs)) are used for this purpose [16]. IGRAs are ex vivo blood tests that measure the T-cell release of IFN-γ following stimulation by antigens specific to the MTBC (with the exception of BCG substrains), i.e., culture filtrate protein 10 (CFP-10) and early secreted antigenic target of 6 kDa protein (ESAT-6).

Lebanon remains a low-TB-burden country, with an estimated incidence of 13 per 100,000 population and a total of 474 notified cases in 2021 [17]. Recently, the estimated prevalence of CKD in Lebanon was 12.5%, which is surprisingly higher than the globally estimated prevalence of 9.1% in 2017. It is worth noting that the burden of dialysis in Lebanon is also among the highest worldwide, with 777 patients per million compared to 410 dialysis patients per million people worldwide [8].

Many studies have confirmed an increased risk (6.9- to 52.5-fold) of TB in dialysis patients compared to the general population [18]. Although several reports investigated the epidemiology of LTBI in HD patients in the Middle East and North Africa (MENA) region [19,20,21], there is a lack of data regarding LTBI prevalence and infection-associated factors among this vulnerable population in Lebanon. In this context, this study, conducted during the COVID-19 pandemic, aimed to determine LTBI prevalence among HD patients in Northern Lebanon and to identify potential factors associated with this infection.

## 2. Materials and Methods

### 2.1. Study Design and Population

This investigation was a multicenter cross-sectional analytic study conducted at two time points in Tripoli, North Lebanon. Between August and October 2020, forty-nine (49) patients attending the dialysis unit at a public hospital (Orange-Nassau Governmental Hospital) were enrolled in the survey. In February 2022, forty-four (44) patients attending the dialysis unit at two private hospitals (Nini Hospital and Dar Al-Chifae Hospital) were added to the survey. Overall, we included 93 patients from the three above-mentioned care centers. The study population corresponds to all eligible HD patients attending the dialysis units within the study period. We excluded patients who (i) were younger than 18 years; (ii) had a history of active TB or had TB symptoms at the time of enrollment; or (iii) were non-Lebanese.

### 2.2. Data and Sample Collection and Laboratory Examination

Data on sociodemographic factors (e.g., age, sex, educational level, and income level), medical records (e.g., BCG vaccine, previous contact with active TB patients, and TST), clinical characteristics of dialysis conditions (e.g., cause of ESRD, HD duration, frequency of HD session per week, and HCV and HBV infections), behavior (e.g., smoking and alcoholism), and the presence of other additional co-morbidities were recorded in the study population. A total of 5 mL of blood was drawn from each participant into two lithium heparin tubes for further laboratory examination.

LTBI was investigated using the QuantiFERON-TB Gold Plus (QFT-Plus) (Qiagen, Germantown, MD) assay, performed according to the manufacturer’s instructions. This assay uses four specialized blood collection tubes: a Nil tube (a negative control to adjust for background IFN-γ), a Mitogen tube (a positive control to confirm baseline immune status), and two antigen tubes, TB1 and TB2, which contain specific M. tuberculosis peptides designed to stimulate both CD4 and CD8 T-cells. TB1 contains peptides from ESAT-6 and CFP-10 mycobacterial antigens to target the cell-mediated responses of CD4 T-cells, while TB2 contains the same CD4 antigenic peptides of TB1 in addition to newly designed peptides that stimulate CD8 T-cells. One milliliter of blood was transferred to each QFT-P tube (Nil, TB1, TB2, and Mitogen). The tubes were labeled, shaken, and incubated at 37 °C ± 1 °C for 16–24 h. They were then centrifuged for 15 min at 2000 to 3000 RCF, and the plasma from each tube was then collected and stored at −20 °C until use. The measurement of IFN-γ via ELISA was subsequently performed, and IFN-γ concentrations were obtained using QFT-Plus Analysis Software, version 2.71.

### 2.3. Statistical Analysis

Statistical analysis was performed using R software (R Core team, version 4.1.0; R Studio, version 1.4.1106) using several packages (e.g., summarytools, DescTools, prettyR, dplyr, and tidyr), and the obtained findings were illustrated using the ggplot2 R package. The categorical data were described as frequencies and associated proportions. The difference between the proportions of LTBI among different categories was initially compared using Fisher’s exact test for categorical covariates and a t-test for continuous covariates. Subsequently, multivariable logistic regression analysis was performed with LTBI as the outcome and age, income level, smoking behavior, and HD due to diabetes mellitus as predictors. We also used a backward stepwise model to identify and confirm the associations of covariates with LTBI. The statistical tests were two-sided, with the type I error set at α = 0.05. The code necessary to replicate the analysis is publicly available (http://doi.org/10.6084/m9.figshare.22043252; accessed on 25 March 2023).

## 3. Results

A total of 93 HD patients, 51 men and 42 women, attending the dialysis units of three hospitals in North Lebanon were enrolled in the study and screened for LTBI. The mean age of the study population was 58.3 ± 12.4 years (age range 26–84 years). Most enrolled patients were from Tripoli (62.4%), came from low-income families (71%), and were overweight (Average Body Mass Index (BMI) = 25.7). Out of all patients, 7.5% had previous contact with a confirmed TB patient, and 39.8% had a history of BCG vaccination. The mean duration of HD was 6.8 ± 4.9 years. Other sociodemographic and clinical characteristics of the study population are summarized in Table 1**.**

The QFT-Plus results showed that out of 93 HD patients, nine had indeterminate QFT-Plus results and were therefore excluded from subsequent statistical analysis. Among the remaining 84 participants with valid QFT-Plus results, 16 tested positive, reflecting a prevalence of 19% (95% interval for *p*: 11.3%, 29.1%). Among patients who had positive QFT-Plus results, the majority were men (62.5%), smokers (56.25%), and BCG-unvaccinated (68.8%) and had low-income status (93.8%). Regarding HD etiologies, high blood pressure and diabetes mellitus were more common among patients with positive QFT-Plus compared to peers with negative results. Multivariable logistic regression analysis showed that LTBI was significantly associated with age [OR = 1.06; 95% CI = 1.01 to 1.13; *p* = 0.03] and a low-income level [OR = 9.29; 95% CI = 1.62 to 178; *p* = 0.04] (Table 2).

## 4. Discussion

People with ESRD have an increased risk of active TB [22]. This dual public health threat commonly affects low- and middle-income countries the most, as the CKD burden is increasing more rapidly in these countries and their resident households are at a greater risk of contracting TB [23,24]. Due to the lack of effective surveillance systems, the official data on ESRD and TB might underestimate the actual rates of these diseases and consequently prevent the implementation of appropriate preventive measures.

Since late 2019, Lebanon has been facing calamitous economic and political crises, which were further exacerbated by the COVID-19 pandemic and the blast of the Beirut port [25]. Many sectors were severely affected, including the healthcare sector, endangering the ability of healthcare centers to provide life-saving care services. Indeed, unprecedented inflation and its consequent health and socioeconomic results have negatively impacted both HD and TB services. Prices of healthcare consultations and life-saving medicines for serious illnesses have increased dramatically, which precipitously threatens the control and treatment of infectious and non-communicable diseases in the Lebanese community [26]. Moreover, laboratory diagnostic supplies and essential drugs have been widely reported as inaccessible to a large part of the Lebanese population [27,28]. In addition to the prevailing extreme poverty and malnutrition (two of the major social determinants of TB) in the Lebanese community, the Beirut port explosion in August 2020 heavily damaged the national TB program (NTP) premises in the Karantina region of the capital Beirut, where the central unit and main TB center are located [17]. The COVID-19 pandemic has also played a key role in increasing TB cases in Lebanon, where TB control services were significantly altered by the pandemic [17]. Furthermore, there was a threat of dialysis service suspension due to severe shortages in supplies. This fact impacts not only HD patients but also individuals who will not be diagnosed and cases who will not be operated on [29].

To the best of our knowledge, we determined the prevalence of LTBI for the first time in the MENA region with the latest generation of IGRAs—the QFT test (i.e., QFT-Plus). Unlike the previous QFT generations (i.e., QuantiFERON-TB Gold In-Tube (QFT-GIT) and QuantiFERON-TB Gold (QFT-G)), the QFT-Plus assay stimulates both CD4 and CD8 T-cells, which makes it very useful in conditions of immune depression due to CD4 T-cell impairments [30]. Indeed, QFT-Plus is the most relevant available tool for appropriately identifying LTBI in individuals with specific immune disorders, including CKD [31].

Our findings show a relatively low prevalence of LTBI (19%) among HD patients. In comparison with other MENA countries (Table 3), Lebanon has the lowest LTBI prevalence among HD patients according to IGRA tests. Higher percentages of LTBI have been reported in Iran [32], Egypt [33], and Saudi Arabia [34], in which 23.4%, 35.1%, and 45.3% of HD patients were shown to have LTBI, respectively. Notably, the highest percentages of LTBI among HD patients were observed in Turkey, ranging from 39.6% to 61% [19,35,36,37,38,39,40]. The low LTBI prevalence found in the present study may be explained by the geographical difference in TB incidence and the fact that Lebanon is a low-TB-burden country [41].

The study population and the sensitivity and specificity of the adopted diagnostic tests may also affect the prevalence of infection. In immunocompromised patients, such as those undergoing HD treatment, the TST performs poorly due to an increased possibility of false-negative results [53]. In addition, Hussein et al. [33] demonstrated that the prevalence of LTBI changed significantly according to the adopted diagnostic method, with 13.5% using the TST and 35.1% using QFT-GIT. The results of most studies in Saudi Arabia [34,47,48], Egypt [33,42], and Turkey [19,36,38,39,40] corroborated that using IGRA tools allows the better detection of LTBI among HD patients. On the other hand, only three studies conducted in Egypt, Iran, and Turkey [32,37,43] showed discordant data, with a higher LTBI prevalence using the TST compared to IGRA tests. To maintain the sensitivity of the QFT-Plus assay in our study population, we collected the samples immediately before the start of the dialysis process. An obvious reduction in IFN-γ production levels in response to TB antigens was detected after the start of the HD process in CKD patients with ESRD [35]; thus, the prior collection of samples was determined to be better to detect LTBI among patients with ESRD.

Patients receiving dialysis are at high risk of developing TB and should be prioritized for LTBI management due to the fact that they present an increased prevalence of MTBC infection and an increased risk of TB reactivation. In this context, current guidelines recommend screening and treating dialysis patients for LTBI [15,54]. In the present study, LTBI was significantly associated with age and a low-income level. A significant association between older age and LTBI was also revealed by Ogawa et al. [55], Shu et al. [56], and Lee et al. [57] in Japan and Taiwan. Low socioeconomic status affects the aspects of both the “exposure-infection-disease-adverse outcome” spectrum and the health of populations in general [58]. On the other hand, other risk factors have been associated with LTBI in other studies. Two studies carried out in Taiwan and Indonesia (Bandiara et al.) showed a significant association between smoking behavior and LTBI [12,56]. Hayuk and colleagues [59] revealed that alcohol consumption was significantly associated with LTBI among HD patients. Moreover, Turkish investigations reported that QFT positivity was more frequent in males compared to females [38] and in patients with previous contact with active TB cases [36].

Our data revealed that 9.7% of HD patients had indeterminate results, which is in agreement with the wide percentage range (2% to 40%) reported in previous studies [36,47,60]. For example, 30% of HD patients had indeterminate QFT-GIT results in Egypt [60]. In contrast, another study using the same diagnostic method showed that only 5.9% of HD patients had indeterminate results [55]. Although an indeterminate QFT-Plus result does not indicate a failed test, it does not provide useful information regarding the likelihood of LTBI. An indeterminate result may be related to immunosuppression associated with older age and underlying diseases or due to some technical errors [61]. The rate of indeterminate results in immunocompromised patients is higher than that observed in immunocompetent individuals [62]. In addition, the indeterminate results determined by QFT-GIT were higher compared with QFT-Plus [62]. Since our samples were transported within the specified time interval and the proper procedure was followed during the specimens’ storage and processing, the possibility of a wrong technical practice is potentially excluded. Taken together, the percentage of indeterminate results in our investigation is probably associated with the immunosuppressed state of HD patients.

It is worth bearing in mind that our samples were collected in the period between 2020 and 2022, during which the world was facing the COVID-19 pandemic. Hemodialysis centers were considered high-risk places for COVID-19 transmission, and HD patients represented a group at risk of infection with severe acute respiratory syndrome coronavirus 2 (SARS-CoV-2) due to impaired immunity. Moreover, due to the lockdown and containment measures taken to combat the pandemic, several patients could not access medical care and consequently missed their HD sessions [63,64,65]. Altogether, these factors reduced the life-saving HD frequency and the weekly duration of HD sessions [63,66] and hindered patients’ access to HD units. We may therefore have missed possible cases of HD patients with different clinical conditions whose enrollment may have affected our findings on LTBI prevalence and its associated risk factors in this vulnerable population.

This study had limitations. Due to the cross-sectional design of this study, we were unable to assess the relationship between CKD/ESRD, HD-associated conditions, and LTBI. In parallel, due to logistic reasons, the follow-up of subjects with indeterminate QFT-Plus results has not been performed (i.e., either repeating the IGRA test with a newly obtained blood specimen or administering a TST). Moreover, the low number of enrolled patients at two time points limits the statistical power for tests and consequently the conclusions that can be drawn. Finally, as previously mentioned, our data were collected during the COVID-19 pandemic, which potentially had an impact on our study findings. The epidemiological significance of our findings remains to be confirmed in future studies.

## 5. Conclusions

To our knowledge, this study represents the first report assessing the prevalence of LTBI in HD patients in Lebanon. Our work showed that 19% of the examined HD patients were positive for LTBI, and infection is significantly associated with age and low-income status. The possibility of TB reactivation with extrapulmonary involvement or spreading must be taken into consideration in order to reduce the morbidity and mortality of TB in these patients. Therefore, there is a paramount need to implement effective TB control strategies among this vulnerable population, with special attention to high-risk patients such as HD and peritoneal dialysis patients. For a better understanding of the local epidemiology of LTBI in Lebanon, further large-scale, nationwide studies are required to explore TB determinants in the community and suggest interventions tackling this issue of global concern.

## Figures and Tables

**Table 1 medicina-59-00654-t001:** Sociodemographic and clinical characteristics of the study population and the prevalence of latent tuberculosis infection.

	Total = 93 Patients	
	*n*	%
QuantiFERON-TB		
Positive	16	17.2
Negative	68	73.1
Indeterminate	9	9.7
Hospital		
Orange-Nassau Governmental Hospital (Public)	49	52.7
Nini Hospital (Private)	31	33.3
Dar Al-Chifae Hospital (Private)	13	14.0
Sex		
Female	42	45.2
Male	51	54.8
Age (mean [SD; min–max])	58.3 [12.4; 26–84 years]	
Body Mass Index (BMI) (mean [SD; min–max])	25.7 [4.0; 16.6–38.8]	
District		
Tripoli	58	62.4
Miniyeh-Danniyeh	13	14.0
Akkar	12	12.9
Zgharta	6	6.5
Koura	4	4.3
Education		
Illiterate	26	28.0
Literate	67	72.0
Income level		
Low	66	71.0
Middle to high	27	29.0
BCG vaccination		
Yes	37	39.8
No	50	53.8
Unknown	6	6.5
Smoking		
Yes	40	43.0
No	53	57.0
Drinking alcohol		
Yes	5	5.4
No	88	94.6
Contact with a tuberculosis patient		
Yes	7	7.5
No	86	92.5
Chronic glomerulonephritis *		
Yes	11	11.8
No	82	88.2
Polycystic kidney disease *		
Yes	5	5.4
No	88	94.6
High blood pressure *		
Yes	29	31.2
No	64	68.8
Diabetes mellitus *		
Yes	12	12.9
No	81	87.1
Nephroangiosclerosis *		
Yes	10	10.8
No	83	89.2
Diabetic nephropathy *		
Yes	14	15.1
No	79	84.9
Kidney transplantation		
Yes	6	6.5
No	87	93.5
Hepatitis B infection		
Yes	2	2.2
No	91	97.8
Hepatitis C infection		
Yes	1	1.1
No	92	98.9
Starting hemodialysis in years ago (mean [SD; min–max])	6.8 [4.9; 1–22 years]	
Duration of hemodialysis sessions		
<4 h	39	41.9
≥4 h	54	58.1
Frequency of weekly hemodialysis sessions		
Up to 2 per week	21	22.6
3 per week	72	77.4

* Suggested cause of hemodialysis. Data are presented as mean [standard deviation (SD); min–max] for the continuous variables and as frequency and percentage for categorical variables.

**Table 2 medicina-59-00654-t002:** Determinants of latent tuberculosis among patients on hemodialysis using univariate analysis and multivariable logistic regression models.

	Univariate Analysis			Multivariable Logistic Regression Models					
				Model 1 ^i^			Model 2 ^ii^		
Categorical variables	%		*p*	adj. OR	95%CI	*p*	adj. OR	95%CI	*p*
Sex									
Female ^1^	16.7								
Male	20.8		0.78						
District									
Tripoli ^1^	22.6								
Outside Tripoli	12.9		0.39						
Education									
Illiterate ^1^	25.0								
Literate	16.7		0.38						
Income level									
Middle to high ^1^	4.2								
Low	25.0		0.03	8.75	1.49–169	0.05	9.29	1.62–178	0.04
BCG vaccination									
Yes ^1^	14.7								
No	23.9		0.40						
Smoking									
Yes ^1^	25.7								
No	14.3		0.26	0.43	0.12–1.44	0.17	0.41	0.12–1.35	0.15
Contact with a tuberculosis patient									
Yes ^1^	28.6								
No	18.1		0.61						
High blood pressure *									
Yes ^1^	25.9								
No	15.8		0.37						
Diabetes mellitus *									
Yes ^1^	36.4								
No	16.4		0.21	0.75	0.17–3.79	0.71			
Kidney transplantation									
Yes	0.0								
No	20.3		0.58						
Duration of hemodialysis sessions									
<4 h	20.0								
≥4 h	18.4		1.00						
Frequency of weekly hemodialysis sessions									
Up to 2 per week	16.7								
3 per week	19.7		1.00						
Continuous variables	Infection	No infection	*p*	adj. OR	95%CI	*p*	adj. OR	95%CI	*p*
Age	64.7	57.6	0.05	1.06	1.01–1.13	0.04	1.06	1.01–1.13	0.03
Body Mass Index (BMI)	25.5	25.8	0.73						
Starting hemodialysis in years ago	5.93	6.97	0.39						

Determinants of latent tuberculosis were predicted using univariate (*t*-test and Fisher’s exact test for continuous and categorical variables, respectively) and multivariable analysis (logistic regression models). ^i^ The variables tested by univariate analysis that had a *p* value < 0.30 were included in Model 1 (multivariable logistic regression analysis). ^ii^ In Model 2, a backward logistic regression model was created including only complete cases. ^1^ Reference group. * Suggested cause of hemodialysis.

**Table 3 medicina-59-00654-t003:** Prevalence of latent tuberculosis infection among hemodialysis patients in the Middle East and North Africa (MENA) region.

Country	Publication Year	Number of Patients	Prevalence (%)		Ref.
			Tuberculin Skin Test	IGRA *^, a, b, c^	
Egypt	2014	40	15	20 ^c^	[42]
	2016	60	45	31.7 ^b^	[43]
	2017	74	13.5	35.1 ^c^	[33]
Iran	2008	100	16	ND ^!^	[44]
	2012	255	20.8	ND	[45]
	2014	47	43.5	23.4 ^b^	[32]
	2022	119	81.5	ND	[21]
Iraq	2016	71	28.6	ND	[46]
Saudi Arabia	2013	133	19	39 ^a^	[47]
	2013	200	13	32.5 ^c^	[48]
	2015	181	17.4	45.3 ^c^	[34]
Tunisia	2002	60	10	ND	[49]
Turkey	2009	50	56	ND	[50]
	2009	56	ND	58.9 ^c^	[35]
	2010	733	38.6	ND	[51]
	2010	100	34	43 ^b^	[36]
	2011	96	43.8	39.6 ^c^	[37]
	2011	89	31.5	45 ^b^	[38]
	2012	411	39	61 ^a^	[39]
	2012	92	30.4	ND	[52]
	2015	50	36.4	54 ^c^	[40]
	2016	95	32	41 ^c^	[19]

***** Interferon-gamma release assay (IGRA) tests performed using T-SPOT.TB ^a^; QuantiFERON-TB Gold (QFT-G) ^b^; or QuantiFERON-TB Gold In-Tube (QFT-GIT) ^c^ method. ^!^ ND, not determined.

## Data Availability

The database and code necessary to replicate the analysis are publicly available (http://doi.org/10.6084/m9.figshare.22043252; accessed on 25 March 2023).

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
