# Peer review of "Prevalence of Latent Tuberculosis Infection among Patients Undergoing Regular Hemodialysis in Disenfranchised Communities: A Multicenter Study during COVID-19 Pandemic"

_medicina, 2023, doi:10.3390/medicina59040654_

Round 1

Reviewer 1 Report

In this article, ''Prevalence of latent tuberculosis infection among patients undergoing regular hemodialysis in disenfranchised communities: A multicenter study during COVID-19 Pandemic'' by Mohamad Bachar Ismai et al, a highlighted some of the major issues (with possible solutions) that the Latin American region is currently dealing with in managing post-COVID-19 pulmonary fibrosis.

In this article, even though including recent and relevant literature, could be structured more clearly and follow a line of thought.

1.      The etiology of kidney disease and the use of medications like (antihypertensives, recombinant human erythropoietin, iron, insulin, diuretics, antidepressants, immunosuppressants, vasodilators, bronchodilators, B vitamins, vitamin C, folic acid, and vitamin D analogs) should be reported and evaluated in terms of their association with TST results.

2.      Patients who failed to return for the reading of the TST results, those who had previously undergone a TST, those who had previously been treated for LTBI, and those who presented with a history of tuberculosis should be excluded from the study. Did the author do?

3.      In a previous study TB setting, deaths due to these diseases were anticipated to increase by up to 10%, 20%, and 36%, respectively over 5 years, in response to the COVID-19 pandemic. The authors should mention it.

4.      The chance of COVID-19 death in patients with TB should described by authors for eg. How many times higher with past TB.  Is TB independently associated with COVID-19 deaths? (regardless of viral loads and degree of immunosuppression?

5.    Did the authors pointed pre-establish criteria for study selection? and also the authors did not use formal tools for critical appraisal of the literature, as associate with the goal of the study. The author should give more explanation in abstract and in the aim of the study about COVID19 pandemic and kidney disease.

Author Response

Dear Reviewer 1,

We are pleased that you found our manuscript interesting, and we thank you for the thoughtful reading and constructive comments.

Please find a revised version of our manuscript. As requested, we answered all your comments and suggestions. All answers are listed below and included in the revised manuscript.

Thank you for considering this revised version of our manuscript.

--------------------------------------------------------------------------------------

Comment 1. In this article, ''Prevalence of latent tuberculosis infection among patients undergoing regular hemodialysis in disenfranchised communities: A multicenter study during COVID-19 Pandemic'' by Mohamad Bachar Ismai et al, a highlighted some of the major issues (with possible solutions) that the Latin American region is currently dealing with in managing post-COVID-19 pulmonary fibrosis.

REPLY: Here we will only mention that the study is not performed in the Latin American region. The study is conducted in Lebanon

Comment 2. The etiology of kidney disease and the use of medications like (antihypertensives, recombinant human erythropoietin, iron, insulin, diuretics, antidepressants, immunosuppressants, vasodilators, bronchodilators, B vitamins, vitamin C, folic acid, and vitamin D analogs) should be reported and evaluated in terms of their association with TST results.

REPLY: Previous data evaluated the etiology of kidney disease and the use of medications evaluated in terms of their association with TST results and showed that some of them (e.g. etiology of CKD, iron store, se of vitamin D analogs, etc..) no significant associations with the results of TST (Ref: Fonseca JC, Caiaffa WT, Abreu MN, Farah Kde P, Carvalho Wda S, Spindola de Miranda S. Prevalence of latent tuberculosis infection and risk of infection in patients with chronic kidney disease undergoing hemodialysis in a referral center in Brazil. J Bras Pneumol. 2013 Mar-Apr;39(2):214-20. doi: 10.1590/s1806-37132013000200013).

However, in this study, LTBI prevalence was evaluated using IGRA instead of TST. and patients were not asked concerning their specific use of each of medications cited above. For these reasons, we did not report and evaluate the etiology of kidney disease and the use of medications in terms of their association with TST results.

Comment 3. Patients who failed to return for the reading of the TST results, those who had previously undergone a TST, those who had previously been treated for LTBI, and those who presented with a history of tuberculosis should be excluded from the study. Did the author do?

REPLY: As cited above, IGRA instead of TST was used in this study. Notably, when available in their medical records, TST data of our participants was obtained and no one from these patients was TST positive. On the other hand, all participants with a history of active TB or having TB symptoms at the time of enrollment were excluded from the study as we cited at the end of the section dealing with “Study design and population” in the Materials and Methods.

Comment 4. In a previous study TB setting, deaths due to these diseases were anticipated to increase by up to 10%, 20%, and 36%, respectively over 5 years, in response to the COVID-19 pandemic. The authors should mention it.

REPLY: In response to the COVID-19 pandemic, it was predicted that deaths due to TB in high TB settings would rise by up to 20% over 5 years (ref: Trajman A, Felker I, Alves LC, Coutinho I, Osman M, Meehan SA, et al. The COVID-19 and TB syndemic: the way forward. Int J Tuberc Lung Dis. 2022;26(8):710-9). The prolonged periods of reduced diagnosis and treatment of new TB cases were predicted to cause the greatest increase in TB-related deaths.

This information, including reference, was added in the first paragraph in the introduction.

Comment 5. The chance of COVID-19 death in patients with TB should described by authors for eg. How many times higher with past TB.  Is TB independently associated with COVID-19 deaths? (Regardless of viral loads and degree of immunosuppression?

REPLY: Limited and conflicting evidence exists regarding the effect of past or current TB on COVID-19 in patients with COVID-19/TB comorbidity. Two large cohort studies conducted in South Africa came to the conclusion that TB is an independent risk factor for increased mortality due to COVID-19 (refs: Jassat W, et al. Risk factors for COVID-19-related in-hospital mortality in a high HIV and tuberculosis prevalence setting in South Africa: a cohort study. Lancet HIV. 2021;8(9):e554–567. doi: 10.1016/S2352-3018(21)00151-X; Risk Factors for Coronavirus Disease 2019 (COVID-19) Death in a Population Cohort Study from the Western Cape Province, South Africa doi: 10.1093/cid/ciaa1198).

Male sex, older age, diabetes, hypertension, and chronic kidney disease were all linked to COVID-19 death. The risk of COVID-19 death was 2.7 times higher in patients with current TB and 1.5 times higher in those with past TB. In addition, the risk of recovery in TB/COVID-19 coinfected patients was 25% lower than in COVID-19 free ones, with a shorter time-to-death (ref.: Sy KTL, Haw NJL, Uy J. Previous and active tuberculosis increases risk of death and prolongs recovery in patients with COVID-19. Infect Dis (Lond). 2020;52(12):902-7).

These data, including references, were added in the first paragraph in the introduction.

Comment 6. Did the authors pointed pre-establish criteria for study selection? and also the authors did not use formal tools for critical appraisal of the literature, as associate with the goal of the study. The author should give more explanation in abstract and in the aim of the study about COVID19 pandemic and kidney disease.

REPLY: Studies used are associated with the goal of our work and were selected on this basis.

In the abstract we clarify that COVID-19 has not only catastrophic effects on TB but it also increased the risk of mortality and hospitalization in HD patients. The effect of COVID-19 on Kidney patients was also discussed in a specific paragraph in the discussion. To Note, at the end of the introduction, we also mention that the study was conducted during the COVID-19 pandemic.

Reviewer 2 Report

 The authors reported the prevalence of LTBI in HD patients in Lebanon. this work showed that 19% of the examined HD patients were positive for LTBI and infection is significantly associated with age and low-income status. The possibility of TB reactivation with extrapulmonary involvement or spreading must be taken into consideration in order to reduce morbidity and mortality of TB in these patients. Therefore, there is a paramount need to implement effective TB control strategies among this vulnerable population, with special attention to high-risk patients such as HD and peritoneal dialysis patients. For a better understanding of the local epidemiology of LTBI in Lebanon, further large- scale nationwide studies are required.

Author Response

We are pleased that you found our manuscript interesting, and we thank you for the thoughtful reading and constructive comments.

Reviewer 3 Report

In the article colleagues have introduced current data.

The following items should be corrected:

-              It is recommended to structure the article, presenting the aim of the study, characteristic of patients in the materials and methods with a description of the design of the study.

-          How was diagnosis of LTBI established? This methodology must be clearly written.

-          It will be very good if authors include the definition of LTBI as well.

All these concerns should be well addressed to consider this manuscript suitable for publication.

Author Response

Dear Reviewer 3,

We are pleased that you found our manuscript interesting, and we thank you for the thoughtful reading and constructive comments.

Please find a revised version of our manuscript. As requested, we answered all your comments and suggestions. All answers are listed below and included in the revised manuscript.

Thank you for considering this revised version of our manuscript.

----------------------------------------------------------------------------------------

Comment 1. It is recommended to structure the article, presenting the aim of the study, characteristic of patients in the materials and methods with a description of the design of the study.

REPLY: Thank you for your valuable suggestion. While we appreciate your input, we would like to clarify that we have already included the characteristics of the patients in the results section of the article. We obtained this information from the questionnaire administered to the participants. As such, we believe it is more appropriate to keep this information in the results section, rather than the materials and methods section. Nonetheless, we will take your comment into consideration for future studies.

Comment 2. How was diagnosis of LTBI established? This methodology must be clearly written.

REPLY: The method used for the diagnosis of LTB was clarified in the materials and methods section. Some info concerning this assay were also added in the introduction.

Comment 3. It will be very good if authors include the definition of LTBI as well.

REPLY: This was done in the introduction.
